# Detection of Larch Forest Stress from Jas's Larch Inchworm (*Erannis jacobsoni* Djak) Attack Using Hyperspectral Remote Sensing

**Guilin Xi** [1,2] , **Xiaojun Huang** [1,3,4,*], **Yaowen Xie** [2], **Bao Gang** [1,3], **Yuhai Bao** [1,3], **Ganbat Dashzebeg** [5], **Tsagaantsooj Nanzad** [5], **Altanchimeg Dorjsuren** [6], **Davaadorj Enkhnasan** [6] and **Mungunkhuyag Ariunaa** [5]

1  College of Geographical Science, Inner Mongolia Normal University, Hohhot 010022, China; xigl21@lzu.edu.cn (G.X.); baogang@imnu.edu.cn (B.G.); baoyuhai@imnu.edu.cn (Y.B.)
2  College of Earth and Environmental Sciences, Lanzhou University, Lanzhou 730000, China; xieyw@lzu.edu.cn
3  Inner Mongolia Key Laboratory of Remote Sensing & Geography Information System, Hohhot 010022, China
4  Inner Mongolia Key Laboratory of Disaster and Ecological Security on the Mongolia Plateau, Hohhot 010022, China
5  Institute of Geography and Geoecology, Mongolian Academy of Sciences, Ulaanbaatar 15170, Mongolia; Ganbat_d@mas.ac.mn (G.D.); tsagaantsoojn@mas.ac.mn (T.N.); mungunkhuyaga@mas.ac.mn (M.A.)
6  Institute of Biology, Mongolian Academy of Sciences, Ulaanbaatar 13330, Mongolia; altanchimeg_d@mas.ac.mn (A.D.); enkhnasand@mas.ac.mn (D.E.)
*  Correspondence: huangxiaojun@imnu.edu.cn

**Abstract:** Detection of forest pest outbreaks can help in controlling outbreaks and provide accurate information for forest management decision-making. Although some needle injuries occur at the beginning of the attack, the appearance of the trees does not change significantly from the condition before the attack. These subtle changes cannot be observed with the naked eye, but usually manifest as small changes in leaf reflectance. Therefore, hyperspectral remote sensing can be used to detect the different stages of pest infection as it offers high-resolution reflectance. Accordingly, this study investigated the response of a larch forest to Jas's Larch Inchworm (*Erannis jacobsoni* Djak) and performed the different infection stages detection and identification using ground hyperspectral data and data on the forest biochemical components (chlorophyll content, fresh weight moisture content and dry weight moisture content). A total of 80 sample trees were selected from the test area, covering the following three stages: before attack, early-stage infection and middle- to late-stage infection. Combined with the Findpeaks-SPA function, the response relationship between biochemical components and spectral continuous wavelet coefficients was analyzed. The support vector machine classification algorithm was used for detection infection. The results showed that there was no significant difference in the biochemical composition between healthy and early-stage samples, but the spectral continuous wavelet coefficients could reflect these subtle changes with varying degrees of sensitivity. The continuous wavelet coefficients corresponding to these stresses may have high potential for infection detection. Meanwhile, the highest overall accuracy of the model based on chlorophyll content, fresh weight moisture content and dry weight moisture content were 90.48%, 85.71% and 90.48% respectively, and the Kappa coefficients were 0.85, 0.79 and 0.86 respectively.

**Keywords:** Jas's Larch Inchworm; continuous wavelet coefficients; chlorophyll content; fresh weight moisture content; dry weight moisture content; SVM

## 1. Introduction

Jas's Larch Inchworm (*Erannis jacobsoni* Djak) is a forest pest that is mainly distributed in northern and northeastern Mongolia. As the main defoliators of coniferous forests in Khentii Province, Mongolia, Jas's Larch Inchworm (JLI) poses serious threats to the ecological security of the Siberian larch (*Larix sibirica Ledeb*) forest [1]. Mongolia is rich in forest resources dominated by coniferous forests, which make up 76.6% of the total

forest area and provide good habitat for insects. According to statistics from the Mongolian Ministry of Forestry, the area of the larch forest threatened by JLI increased from 46,838 hm$^2$ to 292,833 hm$^2$ between 2010 and 2017, making it the most serious pest in Mongolia. Forest destruction will increase the likelihood of forest fires and pose a serious threat to the forest ecosystem [2,3]. According to reports, as of 17 April 2020, Mongolia has 23 counties in seven provinces including Arhangei, Bulgan, Donald, Serenge, Sukhbaatar, Kenti and Huwengur. A total of 31 forest and grassland fires occurred over an area of 9.82 hm$^2$ and caused severe economic damage. With global warming, longer dry spells and longer summers have created conditions for harmful forest pests to survive, which has increased the pressure on forest pests [4,5]. We have checked a lot of the literature on forest pest and found that there are few studies on JLI pests in the world. In particular, there are few experimental studies on remote sensing technology, and it is difficult to control the spread of JLI in larch forests.

Timely detection of pests and diseases is an important link for foresters to control the spread of pests [6]. Traditional JLI detection methods rely heavily on the visual recognition and empirical analysis of local national experts. This method is time-consuming, labor-intensive, and problematic [7–9]. Even in the early stages of the attack, JLI did not show any obvious symptoms on the forest canopy, and it was difficult to be detected in time. Hence, it is very important to develop an effective method for detecting JLI infection in Siberian larch forests. The pest feeds on needle leaves and twigs, altering the content of biochemical components (such as chlorophyll content and water content) in leaves from late May to June (larval stage) every year [10]. As the severity of pest infestation increases, the loss rate of needles will increase, and the color of the Siberian larch forest canopy will change from green ("green attack") to yellow ("yellow attack") to red ("red attack"), and finally to gray ("gray attack") [10]. The transition period from the green canopy to the yellow canopy is called the green attack stage, which is the early stage of pest infestation [11,12]. Many studies on plant diseases and insect pests show that the detection rate of yellow, red and gray attack is higher [13–15]. However, the detection of "green attack" is relatively low, although the stress of biochemical components such as leaf chlorophyll content and water content are obviously detected at this stage [16,17]. The study also showed that hyperspectral techniques can be used to estimate the chlorophyll and water content of forest leaves under pest stress. For instance, RL et al. [18] analyzed the sensitivity between absorption characteristics, three-band ratio indices of spectra, and corresponding relative water content of oak leaves. They concluded that the relative moisture content of oak leaves and the absorption characteristic parameters exhibit linear relationships at 975 nm, 1200 nm, and 1750 nm, which indicates that hyperspectral can capture changes in moisture content. Zhang et al. [19] used spectral continuous wavelet coefficients to estimate the chlorophyll content of corn pests, and the results confirmed the potential of hyperspectral inversion for determining chlorophyll content. Asner et al. [20] developed a spectrum feature of Rapid Ohia Death (ROD) and found that 80% of plants infected with fungal pathogens had reduced water content and chlorophyll content. From what has been said above it can be seen that hyperspectral data are obviously very sensitive to these small changes and can offer technical assistance in the detection of pest forests.

The hyperspectral continuous wavelet transform detects small changes in forest tree infestation status, which can improve the weak or insignificant spectral signals caused by the infestation and highlight some characteristic information, such as the spectral absorption and reflection properties of the shape and position of the canopy or leaves [21–23] Therefore, the use of continuous wavelet coefficients is very important for detecting forest pests at different stages of infestation. This experiment aims to explore a new method for extracting continuous wavelet spectral features sensitive to chlorophyll and moisture content. Pearson correlation analysis is usually used to extract sensitive spectral features [19,24–26]. Then the more sensitive spectral bands are further screened out by statistical analysis. For the correlation analysis of the entire waveband, a correlation value is obtained for each waveband, so that the obtained value is continuous and forms a waveband curve with inconsistent fluctuations. Peck et al. [27] A review article summarized some peak extraction

algorithms, from which we found that the Findpeaks function can effectively divide the highest peaks into an interval. This is an effective algorithm for extracting sensitive bands and is worth using in this research. Since the sensitive band extracted by the Findpeaks function may not correctly explain the contribution of the relevant band to target detection, we introduce a successive projections (SPA) algorithm. If the number of original bands is large, the process will take a long time, but the RMSE value can quantify the whole process. This feature makes SPA more suitable for practical applications [28].

In addition, some machine learning algorithms have been successfully applied to detect the symptoms of plant infestation. For example, Tian et al. [29] used the SVM algorithm to successfully identify the different degrees of damage to rice leaf blast infected rice plants, and the classification accuracy rate in both the asymptomatic phase and the early stage of infection exceeded 80%. Sarangdhar et al. [30] developed a support vector machine algorithm to identify cotton leaf diseases, and the classification result was 83.26%. These studies show that the support vector machine algorithm has great potential in monitoring plant symptoms. Support Vector Machine is a non-parametric method that attempt to use an optimal hyperplane for training data in a multidimensional feature space. Therefore, when data is classified from multiple sources, it can have better classification accuracy than non-parametric methods such as the method of maximum likelihood. At present, the remote sensing research using hyperspectral continuous wavelet coefficients to detect pest infection status is still immature. In particular, the hyperspectral classification of asymptomatic, early and infected larch has not, to our knowledge, been attempted under the stress of JLI infestation. In response to the above problems, the overall goal of this research is to establish a classification method based on Findpeaks-SPA-SVM to provide the most important experimental data and theoretical basis for large-scale detection of JLI, identification of asymptomatic, early and infected larch trees. Therefore, the specific objectives are: (1) By analyzing the differences in the sensitivity of different biochemical components of larch to different hyperspectral continuous wavelet coefficients, to explore the potential of different biochemical components in detecting pest infected forests; (2) Evaluate the classification accuracy of Findpeak-SPA-SVM algorithm in asymptomatic, early and infection stages and the best combination of hyperspectral features.

## 2. Materials and Methods

### 2.1. Study Area

The study site is a larch forest in northeastern Mongolia (110°46′1.2″E to 110°46′33.6″E, 48°26′13.2″N to 48°26′34.8″N) (Figure 1). It is located 100 km from Khentii Province and has a total area of 16.75 hm$^2$ and an annual average altitude of 1330 m. The study area has a continental climate, with an annual average temperature of 20 °C in June and July and an average annual precipitation of 200–300 mm [31]. The forest has a single tree species (larch), and it is characterized by poor forest quality and susceptibility to outbreaks of JLI. Larch trees in the study area exhibit different levels of damage, which can represent different stages of pest infestation. Therefore, it can meet the needs of this research.

### 2.2. Data Preparation and Preprocessing

#### 2.2.1. Selection of Sample Trees

JLI pests damage larch forests mainly during the period from late May to mid-July (larval stage). During this time, the larvae will eat a large number of healthy needles and twigs, which will seriously damage the tree and change the color of the leaves [1,10]. Therefore, sample trees were selected on the basis of leaf loss rate and canopy color data. Canopy color data were determined through a combination of visual discrimination in the field and indoor photo identification (using the straw tool of Adobe Photoshop software to obtain the RGB information of larch canopy photos) (Figure 2). For the data of leaf loss rate, typical branches were selected from the upper, middle, and lower levels of each sample tree, and a total of six typical branches were selected. Then the number of damaged needles and

healthy needles were counted, and the leaf loss rate of each sample number was calculated using Equation (1):

$$LLR_i = \frac{1}{6} \sum_{i=1}^{6} \frac{N_{Li}}{(N_{Li} + N_{Hi})} \times 100\%,$$ (1)

where $LLR_i$ is the leaf loss rate of the $i$-th ($i = 1, 2, 3 \ldots 79, 80$) sample tree, and its value is in the range of 0% to 100%. $N_{Hi}$ and $N_{Li}$ are the number of healthy needles and damaged needles, respectively, of the $i$-th sample branch. This sampling approach provides a fairly accurate and feasible method for spectral identification of forest canopy affected by insects. The JLI attack is highly likely to begin at the upper crown, and the entire sampling process from the upper to the middle and lower layers is a scalable sampling method for investigating chaotic insect invasion.

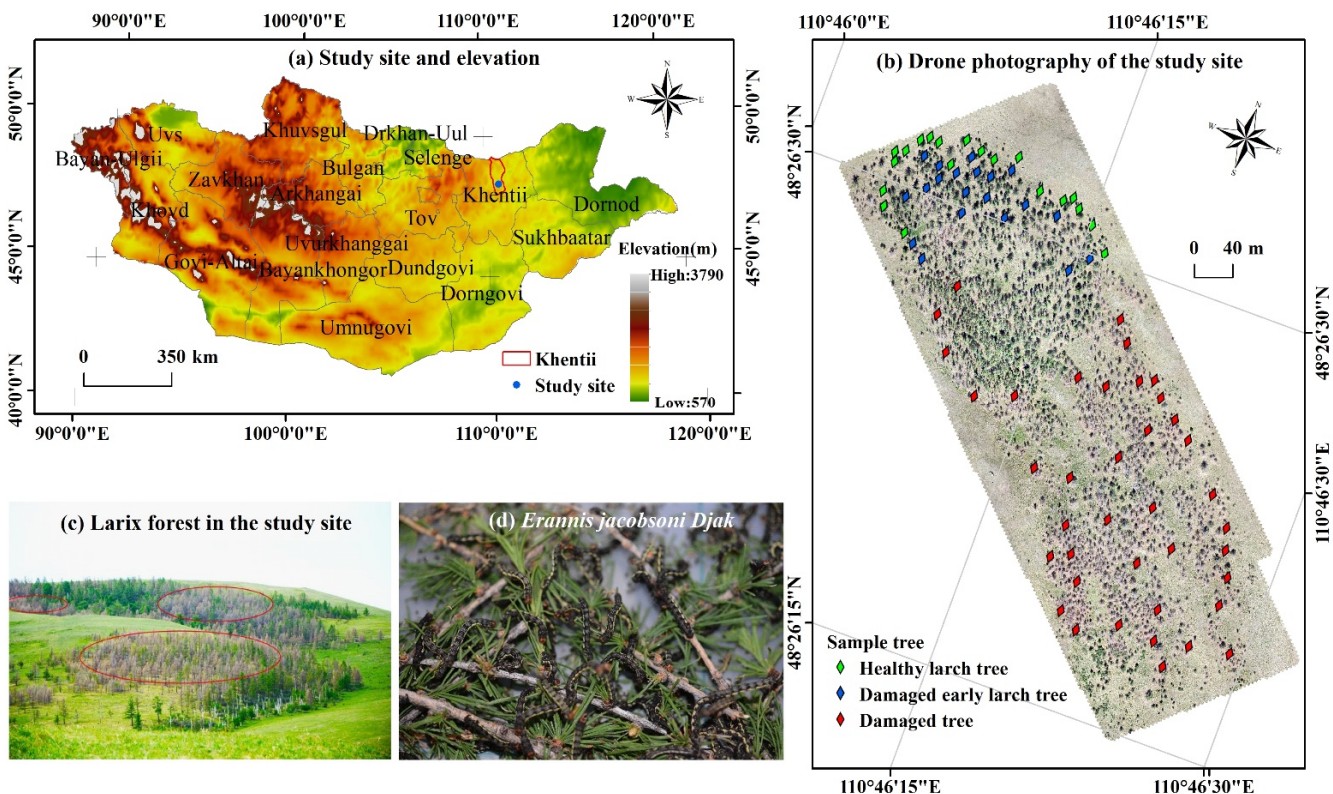

**Figure 1.** (**a**) Study area in Mongolia: Elevation (**b**) Drone photography of the study site; distribution of sample tree (**c**) larch forest in the study area; (**d**) Jas's Larch Inchworm (*Erannis jacobsoni* Djak).

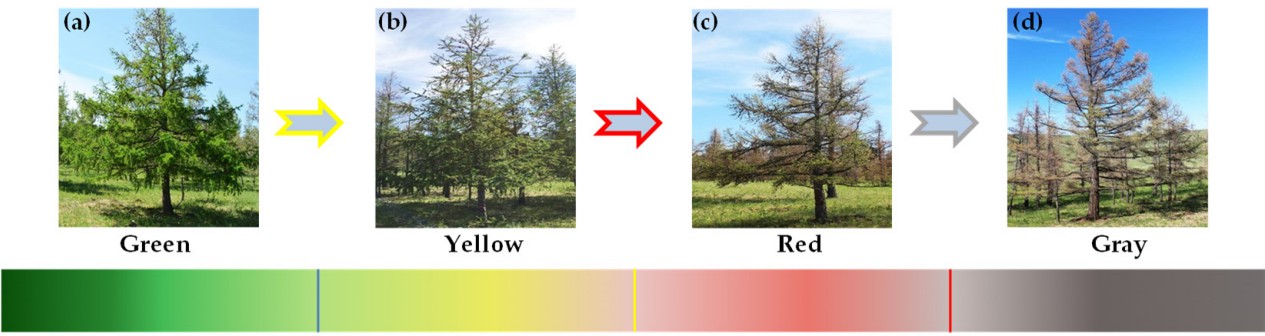

**Figure 2.** Stages of larch canopy: (**a**) Green (**b**) Yellow (**c**) Red (**d**) Gray.

As shown Table 1, sample larch trees with a leaf loss rate of 0–5% and a green canopy color were defined as healthy trees (before attack). Sample larch trees with a leaf loss rate of

5–15% and a green canopy color were defined as damaged early trees (early–stage infection). Sample larch trees with a leaf loss rate of 15–100% and yellow, red, and gray canopy colors were defined as damaged trees (middle to late-stage infection). In this manner, 19 healthy trees, 21 damaged early trees, and 40 damaged trees were selected from the test area as the basic data of the hyperspectral identification model (Table 1).

**Table 1.** Sample tree evaluation table.

| Tree Class | Healthy Tree | Damaged Early Tree | Damaged Tree |
|---|---|---|---|
| Leaf loss rate | 0–5% | 5–15% | 15–100% |
| Canopy color | Green | Green | Yellow, red, and gray |

### 2.2.2. Hyperspectral Data Collection and Preprocessing

In the experiment, the ground object spectrometer, SVC HR-102, was used; this spectrometer has a spectral range of 350–2500 nm. During the collection of hyperspectral reflectance data, the same distance (approximately 20 cm) was maintained between the instrument probe and the needle of the sample tree and the correction distance of the whiteboard; the field of view angle was 25°, vertical downward, and the data were collected under fine weather conditions from 10:30 a.m. to 14:30 p.m. Cut a typical branch from the upper, middle, and lower layers of each sample canopy. Hyperspectral data for each layer were collected five times, and the whiteboard was calibrated for each sample tree to ensure accuracy and reliability of the data.

Regarding the preprocessing of the collected hyperspectral data, random noise interference was first removed using the ViewSpecPro hyperspectral processing software, abnormal and duplicate spectra were then eliminated, and the effective spectra of each sample tree were averaged to obtain the original spectral data representing the sample tree. On the basis of obtaining the original spectral reflection data, the Spectral Math tool of ENVI software was used to obtain the smooth spectral curve (weighted coefficient 5), and then the spectra were transformed into a series of continuous wavelet coefficients on the $2^{1-10}$ scale in MATLAB2021b software. The principle is to transform the original hyperspectral reflectance into a series of wavelet energy coefficients at different scales and positions using wavelet bases. The specific Equations (2) and (3) are as follows:

$$\psi_{a,b}(\lambda) = \frac{1}{\sqrt{a}} \psi\left(\frac{\lambda - b}{a}\right), \tag{2}$$

$$W_f(a,b) = (f, \psi_{a,b}) = \int_{-\infty}^{+\infty} f(\lambda)\psi_{a,b}(\lambda)d\lambda, \tag{3}$$

where $\psi_{a,b}(\lambda)$ is the mother wavelet basis, a and b are the scale factor and the shift factor, respectively, $W_f(a,b)$ is the wavelet coefficient, $f(\lambda)$ is the reflection spectrum, and $\lambda$ is the wavelength (350–1800 nm). The specific mother wavelet bases are as follows: ① Biorthogonal wavelet function series: bior1.3, bior1.5, bior2.2, bior2.4, bior2.6, bior2.8, bior3.1, bior3.3, bior3.5, bior3.7, bior3.9, bior4.4, bior5.5, and bior6.8; ② Coiflets wavelet function series: coif1, coif2, coif3, coif4, and coif5; ③ Daubechies wavelet function series: db1, db2, db3, db4, db5, db6, db7, db8, db9, and db10; ④ Symlets wavelet function series: sym2, sym3, sym4, sym5, sym6, sym7, and sym8.

### 2.2.3. Data Collection and Preprocessing of Biochemical Components

(1)  Chlorophyll content data

The SPAD-502 portable chlorophyll meter was used to measure the relative chlorophyll content of larch needles. In the determination of leaf SPAD value, as leaf thickness, development stage, and environmental conditions are variable, the actual chlorophyll content has a certain influence, and it needs to be strictly controlled [32]. In this study,

only one test area was considered, and its development stage and environmental stage were relatively consistent. Therefore, in order to improve the universality of experimental data, we ensured that the thickness of the needles of each sample tree was as consistent as possible during the clamping of experimental instruments. The specific approach is to ensure that the needle samples to be tested completely cover the instrument receiving window, and the needles should be arranged in a layer. According to this experimental process, at least three repeated measurements were performed on each tree, and the average value was taken to represent the relative chlorophyll content of the sample tree. Then the absolute chlorophyll content was further calculated by Equation (4):

$$y = 0.996x + 1.52, \tag{4}$$

where y is absolute chlorophyll content ($\mu$g/cm2), and x is relative chlorophyll content.

(2)　　Water content data

First, three twigs of similar sizes were cut from each sample tree, and their weights were averaged in the field. Then, they were sealed and taken to the laboratory for drying and weighing. The average weight of branches weighed in the field is called fresh weight, and the average weight of branches dried in the laboratory is called dry weight. It should be noted that the fresh weight and dry weight data cannot truly represent the moisture content of each sample tree. Many scholars use the fresh weight moisture content and dry weight moisture content to express the moisture content of plant tissues [33–35]. The fresh weight and dry weight moisture contents were calculated by Equations (5) and (6):

$$\mathrm{LWCF} \ = \ \frac{\mathrm{FW} \ - \ \mathrm{DW}}{\mathrm{FW}} \times 100\%, \tag{5}$$

$$\mathrm{LWCD} \ = \ \frac{\mathrm{FW} \ - \ \mathrm{DW}}{\mathrm{DW}} \times 100\%, \tag{6}$$

where FW is the fresh weight, DW is the dry weight, and LWCF and LWCD are the fresh weight and dry weight moisture contents of twigs, respectively.

*2.3. Method*

The overall research technical route of the hyperspectral identification of JLI infection forest in the different stage is shown in Figure 3

2.3.1. Sensitivity Analysis

Before conducting a sensitivity analysis, the significant differences in the biochemical composition of healthy, early damaged and damaged trees should be assessed. We calculated the mean, maximum, minimum, and standard deviation (std) of various sample trees (Table 2). The conditions for measuring biochemical composition in the field have some accumulated errors in the process of environmental, instrument and human operation. Therefore, the abnormal and non-compliant sample trees are eliminated. In addition, the experiment considers the authenticity of the measured biochemical composition values, so the data quality of the sample tree we select must ensure that it meets the experimental requirements.

The Pearson correlation analysis method was used to determined significant correlations between the biochemical components and continuous wavelet coefficients. The value of $p$ and the correlation coefficient (r) determine the correlation between them. The smaller the $p$-value, the more significant the correlation between variables. The closer the value of r is to 1, the better the sensitivity. This study analyzed the correlation between forest biochemical composition and spectral continuous wavelet coefficients to understand the distribution of sensitive spectral bands under JLI attack.

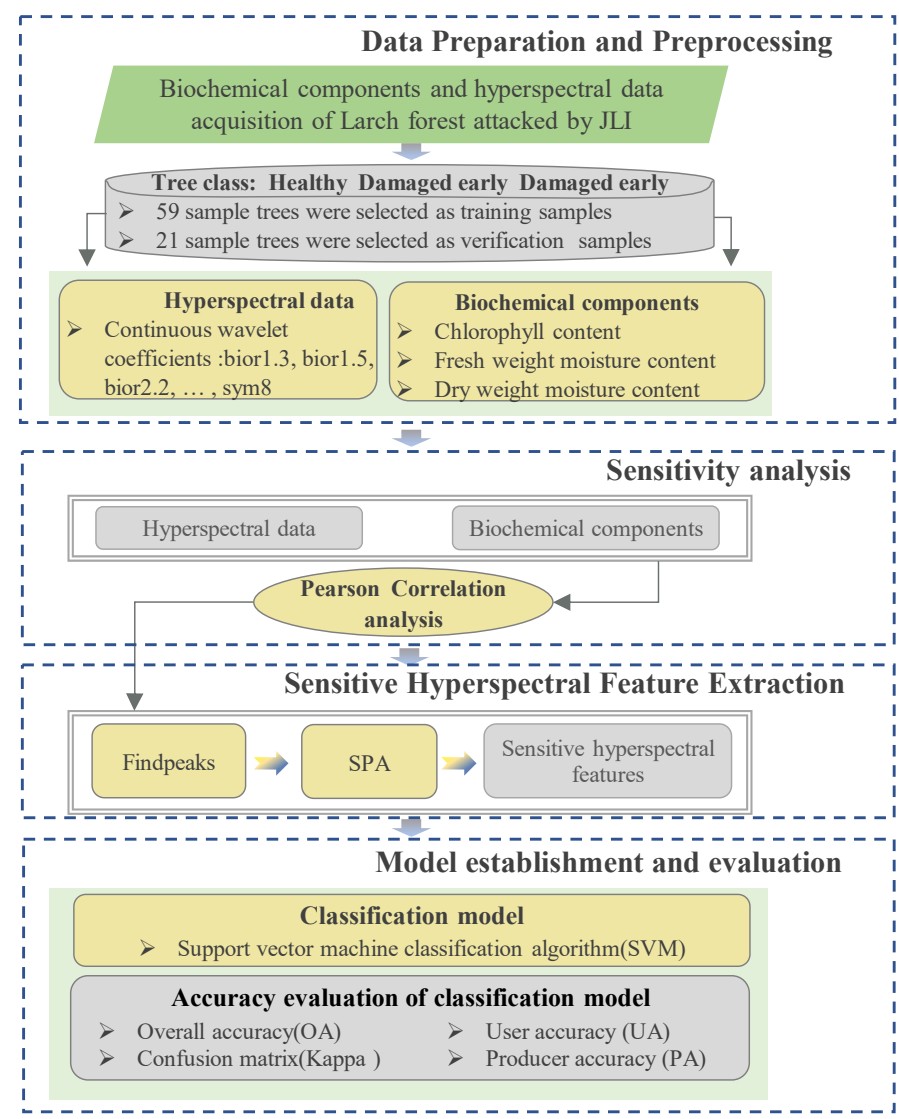

**Figure 3.** Overall research technical route.

**Table 2.** Descriptive statistics of biochemical components of different types of sample trees.

| | Chlorophyll Content (µg/cm²) | | | | Fresh Weight Moisture Content | | | | Dry Weight Moisture Content | | | |
|---|---|---|---|---|---|---|---|---|---|---|---|---|
| | Max | Min | Mean | Std | Max | Min | Mean | Std | Max | Min | Mean | Std |
| Healthy tree | 50.30 | 39.39 | 44.51 | 2.79 | 0.74 | 0.62 | 0.69 | 0.04 | 2.76 | 1.61 | 2.00 | 0.37 |
| Damaged early tree | 38.45 | 28.41 | 32.87 | 3.35 | 0.67 | 0.57 | 0.62 | 0.04 | 1.54 | 1.09 | 1.31 | 0.15 |
| Damaged tree | 27.16 | 4.41 | 13.95 | 5.38 | 0.57 | 0.02 | 0.26 | 0.15 | 1.08 | 0.02 | 0.40 | 0.28 |

### 2.3.2. Sensitive Hyperspectral Feature Extraction

The number of peak points in the waveform function and the corresponding peak values can be rapidly extracted using the Findpeaks function in MATLAB2021b [36–38]. When the value of a function is greater than the left and right adjacent dependent variables, it is defined as a peak value. In this study, the Findpeaks toolbox function was used to automatically find the $R^2$ peak between the biochemical components and hyperspectral features, and then extract the corresponding sensitive hyperspectral features. Findpeaks function has two important parameters. The first is mpd, which is expressed as the minimum distance between adjacent peaks in the frequency band $R^2$; the other is mph, which is expressed as the minimum height between adjacent $R^2$ peaks. There is no clear

standard for the choice of mpd and mph, which depends on the requirements of the experiment. After observing the $R^2$ results and considering the band range, we select the mpd and mph values to be 20 and 0.25, respectively. The result of this parameter selection is to control the number of extracted bands between 0–90 (band range is 350–1800 nm) and its $R^2$ value will be greater than 0.25 ($p < 0.05$) for the purpose of extracting to reach sensitive hyperspectral feature extraction.

The successive projection algorithm (SPA) has been successfully applied in many studies on the dimensionality reduction processing of vegetation hyperspectral features [39–42]. SPA can overcome the collinearity between sensitive bands, select important wavelengths, and establish reliable models. The principle for the SPA to select the sensitive band is that the selected sensitive band is the new sensitive band among all remaining sensitive band and the new sensitive band has the largest projection value on the orthogonal subspace of the previously selected sensitive band and use the root mean square error (RMSE) as a scoring standard to determine the optimal band. The principle and steps of SPA algorithm are explained in many related articles, so there is no more description here. Because of the Findpeaks function extracts many sensitive hyperspectral features, including a large amount of low sensitivity spectral feature information. Therefore, in order to improve the stability and accuracy of the detection model, the SPA is used to further process the Findpeaks processing results to obtain high-sensitivity hyperspectral features (the combined algorithm is denoted as Findpeaks-SPA).

### 2.3.3. Model Establishment and Evaluation

From the data of all larch sample trees, 59 trees (70%) were selected as the training sample data, and the remaining 21 (30%) trees as the verification sample data. The training sample data includes 14 healthy trees, 15 damaged early trees, 29 damaged trees, and the rest is verification sample data. The sensitive spectral features of chlorophyll content, fresh weight moisture content, and dry weight moisture content extracted based on Findpeaks-SPA were taken as independent variables of the model, and sample trees at different stages of damage were taken as the dependent variables. Using the support vector machine (SVM) algorithm, the hyperspectral recognition model was established, and we evaluated the accuracy of the model. SVM algorithm have been widely used in many fields, and I have made many achievements in various applications [43–46]. Therefore, it is worthwhile to try to apply these methods in this research.

Support vector machines are non-parametric supervised classifiers. It follows the strategy of minimizing structural risk, constructs an optimal separation hyperplane, and maximizes the boundary between classes with fewer support vectors. Compared with traditional training methods, this can achieve accurate classification results in a data structure with fewer training samples and stronger aggregation. [47–49]. We use a free library (LibSVM) with radial basis function (RBF) kernel to perform support vector machine tasks. Its classification accuracy and precision are mainly controlled by the parameters c and $\gamma$. $\gamma$ controls the width of the Gaussian kernel and c controls the penalty for training samples on the wrong side of the decision limit. The value of c will determine the number of support vector machines obtained. For example, the smaller the value of c, the smaller the number of support vectors obtained and the greater the classification error; otherwise, the larger the number of support vectors, the problem of overfitting arises. Taking into account the rationality of the parameters c and $\gamma$, we use the libSVM library for grid search and five cross-validations.

In order to evaluate the accuracy of the classification model, we use MATLAB2021b software to construct a confusion matrix for the classification results. We can obtain four model evaluation indexes: user accuracy (UA), producer accuracy (PA), overall accuracy (OA) and Kappa coefficient. The specific effects of these indicators have been explained in many articles [50,51], so they will not be repeated here.

## 3. Results

### 3.1. Sensitivity Analysis of Hyperspectral Features to Biochemical Components

Given that the spectral continuous wavelet coefficients appear to be related to the outbreak of JLI, the correlation coefficient r squared ($R^2$) between spectral continuous wavelet coefficients and the biochemical components (chlorophyll content, fresh weight moisture content and dry weight moisture content) were analyzed. Differences in the responses of the biochemical components to different wavelengths of spectral continuous wavelet coefficients were identified. A significant correlation was observed when the pest outbreak conditions significantly influenced the biochemical components, indicating that the pest outbreak causes varying degrees of changes in the biochemical components of the larch. Therefore, the components exhibited different sensitivities to different bands of the continuous wavelet coefficients of the spectrum. The coefficients between the continuous wavelet coefficients and chlorophyll content, fresh weight moisture content, and dry weight moisture content at each wavelength are shown in Figure 4.

The continuous wavelet coefficients exhibited varying degrees of sensitivity to the biochemical components, although the sensitivities were very similar with subtle differences. In relative terms, the spectral continuous wavelet coefficients were highly sensitive to fresh weight moisture content, followed by chlorophyll content and dry weight moisture content, which indicates that the pest outbreak had a stronger effect on fresh weight moisture content than on chlorophyll content and dry weight moisture content. This finding suggests that the continuous wavelet coefficients can capture these subtle changes, thus facilitating the early identification of pest outbreaks.

Figure 4a shows that the $R^2$ between the continuous wavelet coefficients and chlorophyll content varies with wavelength. From the $R^2$ of different bands, the chlorophyll content appears to have an obviously significant correlation ($R^2 > 0.56$; $p < 10^{-15}$), with good sensitivity mainly in the ranges of 540–581 nm, 596–699 nm, 723–804 nm, 954–956 nm, 974–1011 nm, 1134–1143 nm, 1166–1199 nm, 1245–1275 nm, 1313–1386 nm, 1412–1467 nm, 1483–1485 nm, 1500–1664 nm, 1743–1770 nm, and 1784–1785 nm.

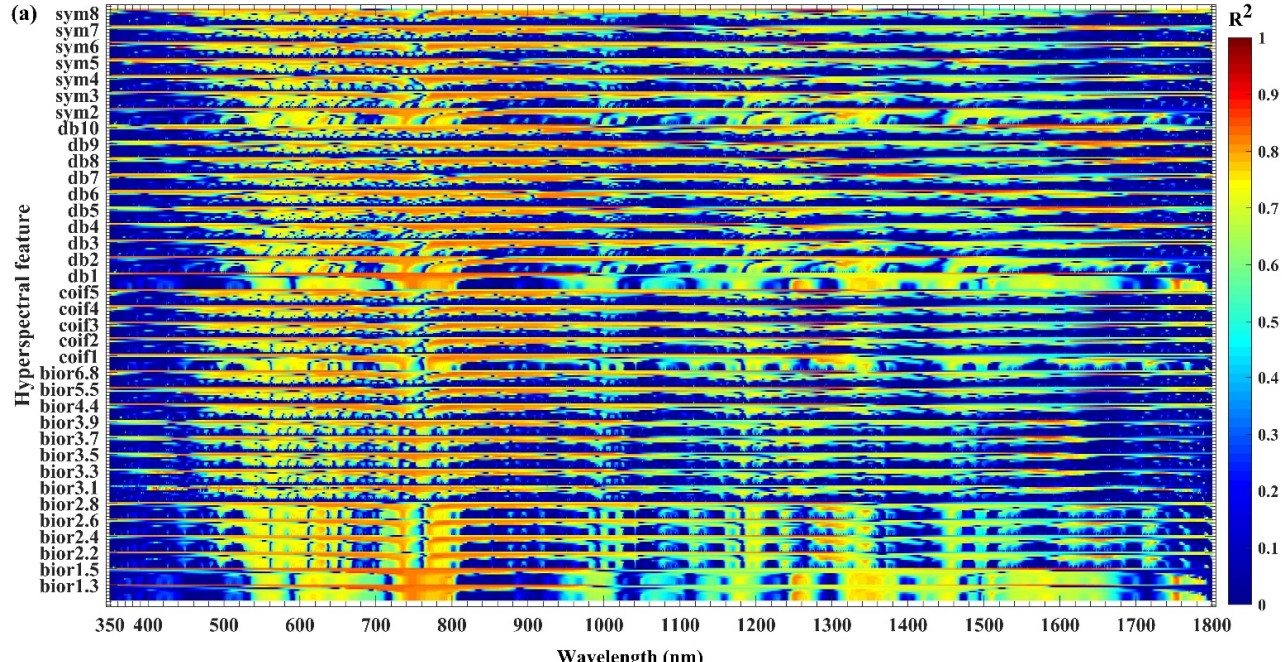

**Figure 4.** *Cont.*

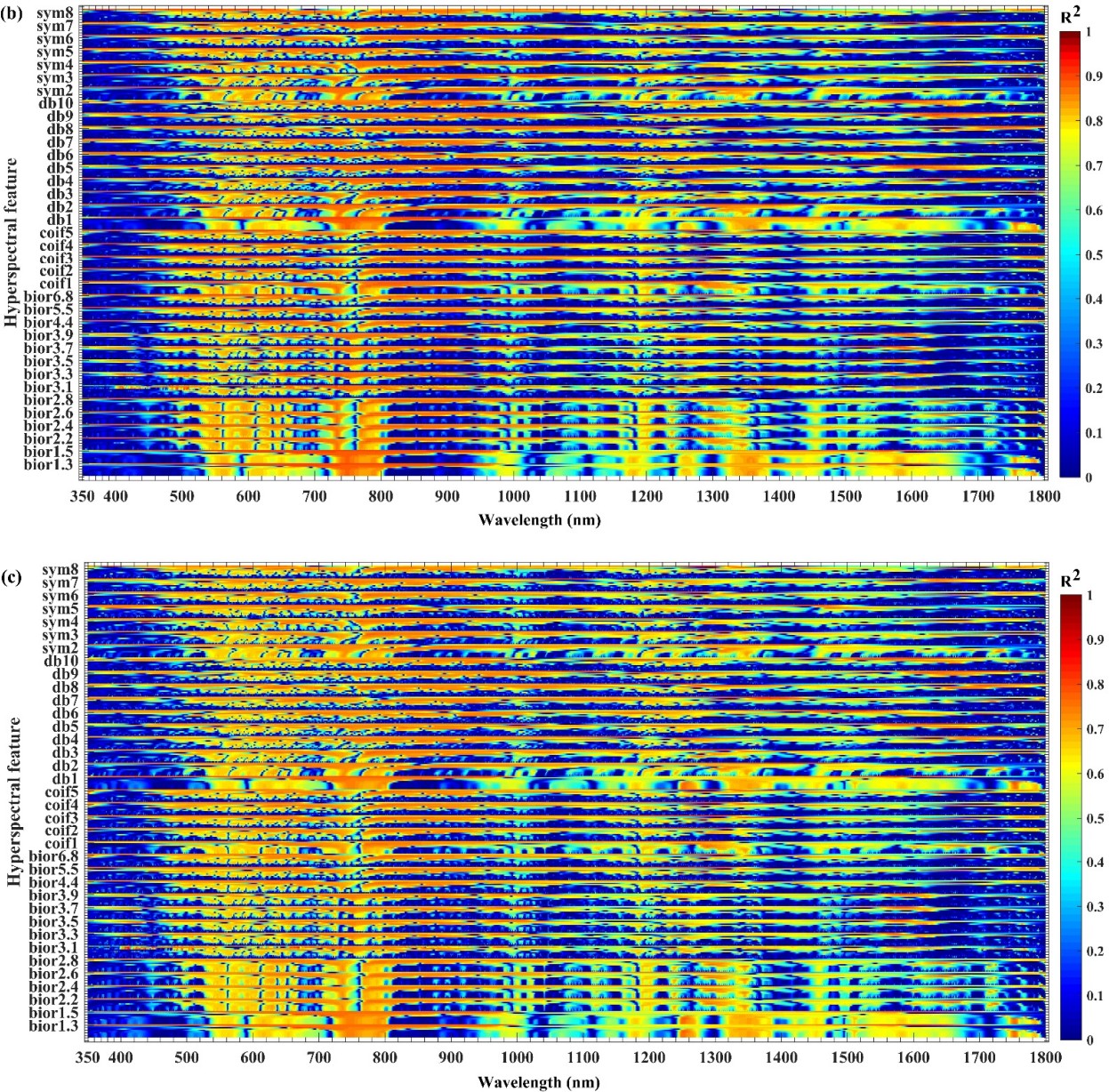

**Figure 4.** Variations of $R^2$ (the correlation coefficient r squared) with wavelength between the spectral continuous wavelet coefficients and: (**a**) chlorophyll content (**b**) fresh weight moisture content; and (**c**) dry weight moisture content.

Figure 4b shows that the $R^2$ between the continuous wavelet coefficients and fresh weight moisture content varies with wavelength. From the $R^2$ of different bands, the fresh weight moisture content appears to have an obviously significant correlation ($R^2 > 0.56$; $p < 10^{-15}$), with good sensitivity mainly in the ranges of 537–583 nm, 595–709 nm, 723–804 nm, 953–962 nm, 967–1010 nm, 1053–1083 nm, 1130–1139 nm, 1162–1206 nm, 1247–1279 nm, 1315–1386 nm, 1404–1468 nm, 1480–1486 nm, 1503–1672 nm, 1746–1772 nm, and 1783–1785 nm.

Figure 4c shows that the $R^2$ between the continuous wavelet coefficients and dry weight moisture content varies with wavelength. From the $R^2$ of different bands, the dry weight moisture content appears to have an obviously significant correlation ($R^2 > 0.56$; $p < 10^{-15}$), with good sensitivity mainly in the ranges of 542–579 nm, 598–679 nm, 723–803 nm, 978–1007 nm, 1168–1195 nm, 1245–1274 nm, 1316–1386 nm, 1416–1426 nm, 1439–1465 nm, 1503–1519 nm, 1533–1654 nm, 1744–1769 nm, and 1784–1785 nm.

The results show that continuous wavelet coefficients can capture the spectral absorption and reflection characteristics caused by chlorophyll and moisture. With increasing degree of damage by the JLI, the rate of leaf loss will gradually increase. At the same time, the chlorophyll content and water content will gradually decrease, leading to obvious responses of the spectral reflectance of the forest canopy [52]. Therefore, the continuous wavelet coefficient of the spectrum and the chlorophyll content, fresh weight moisture content, and dry moisture content of the needles have high potential for application to the detection of JLI outbreak.

### 3.2. Extraction of Sensitive Hyperspectral Feature Bands

In this study, the Findpeaks function and combined Findpeaks—SPA function was used to process the sensitivity function between the continuous wavelet coefficients and the biochemical components and extract sensitive hyperspectral feature bands (Figure 5). As shown in Figure 5a, sensitive hyperspectral features corresponding to the chlorophyll content are mainly in the ranges of 360–522 nm, 772–972 nm, 1140–1433 nm, and 1536–1791 nm. This shows that the blue absorption band (403 nm, 427 nm, 434 nm, 450 nm, 453 nm, 471, and 484 nm) and red absorption band (614 nm and 654 nm) and green reflection peaks (506 nm and 542 nm) of chlorophyll content were well captured using the Findpeaks-SPA function. Figure 5b,c show that sensitive hyperspectral characteristic bands corresponding to fresh weight moisture content are mainly in the ranges of 953–962 nm, 967–1010 and 1404–1468 nm. Sensitive hyperspectral bands corresponding to dry weight water content are mainly in the ranges of 978–1007 nm, 1168–1195 nm, and 1439–1519 nm. This indicates that the Findpeaks-SPA function well captures the absorption bands of water (964 nm, 983 nm, 991 nm, 1099 nm, 1184 nm, 1421 nm, 1493 nm, and 1502 nm). In general, the continuous wavelet coefficients in the bands of 360–522 nm, 772–1010 nm, 1140–1433 nm, 1404–1519 nm and 1536–1791 nm were highly sensitive to the damage degree of larch. This can be explained by the serious decrease in chlorophyll and water content in the conifer of larch forest due to the JLI attack (Table 2). The 360–522 nm band mainly reflects the reflection characteristics of chlorophyll, while the 772–1010 nm, 1440–1433 nm, 1404–1519 nm and 1536–1791 nm bands mainly show the characteristics of water absorption in needles. With the increase of leaf loss rate caused by JLI and the change of larch canopy color from green to yellow to red to gray, the chlorophyll content and the water content of conifers gradually decreased, leading to the gradual increase of spectral reflectance. Therefore, the obtained experimental results are undoubtedly in line with the biological characteristics of plants. These results satisfactorily prove that the Findpeaks-SPA function can effectively extract sensitive hyperspectral features of chlorophyll content and water content.

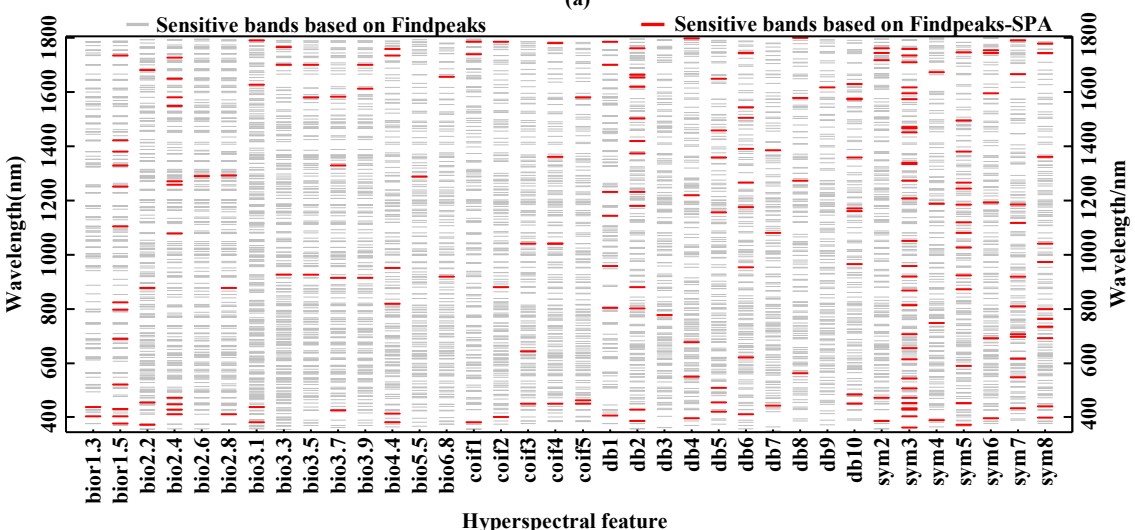

**Figure 5.** *Cont.*

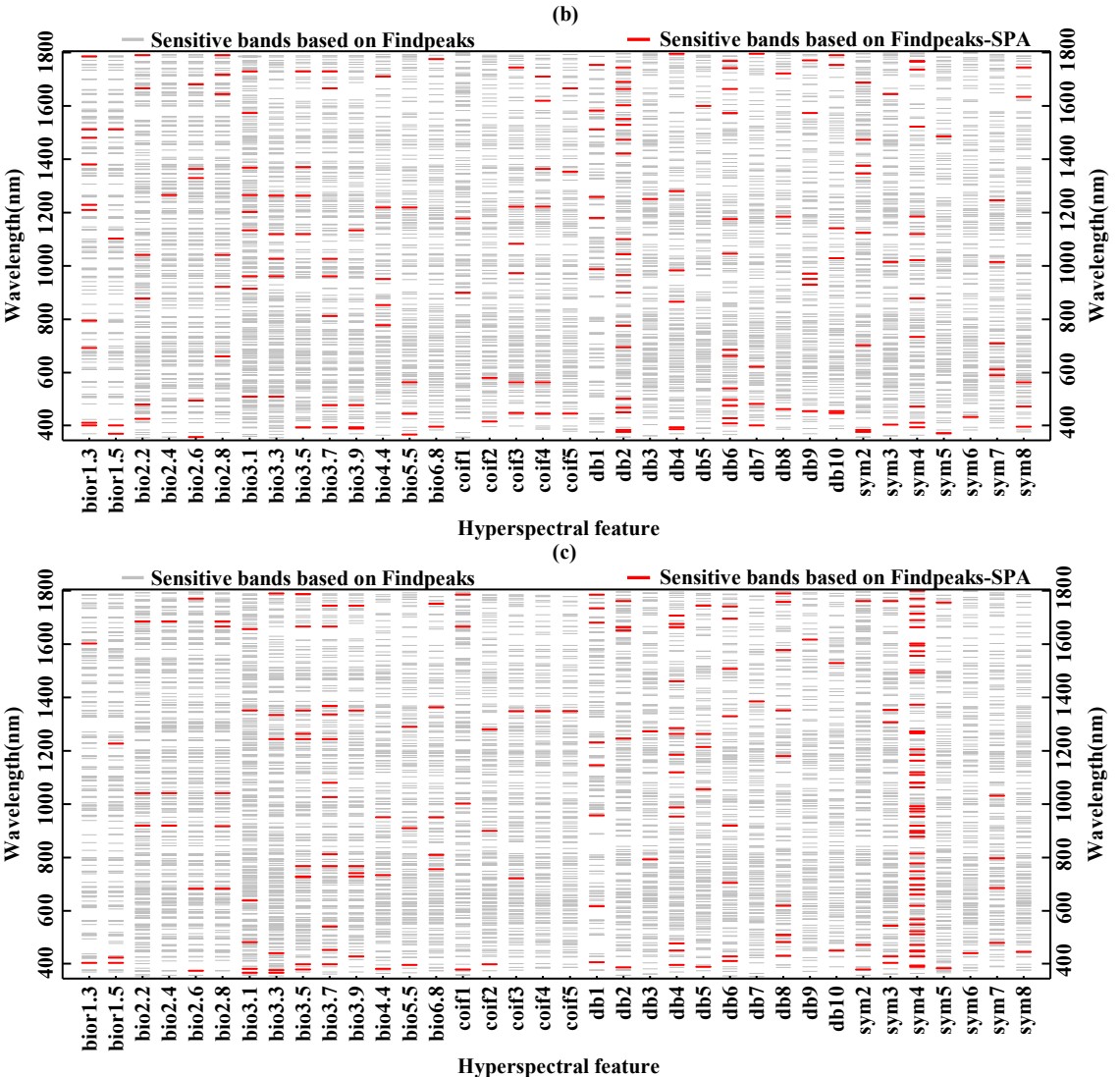

**Figure 5.** Sensitive spectral features: (**a**) chlorophyll content; (**b**) fresh weight moisture content; (**c**) dry weight moisture content.

*3.3. Model Results*

Figure 6 shows the overall accuracy (OA) and kappa coefficient (A) of the SVM classifier for chlorophyll content, moisture content in fresh weight, and moisture content in dry weight of different trees. Chlorophyll content shows high recognition accuracy on bior2.8, coif3 and sym3 (OA: 0.81–0.90; K: 0.72–0.85), and the accuracy of these mother wavelet bases is higher than that of fresh weight moisture content and dry weight moisture content except that OA and K (0.81, 0.74) similar to dry weight moisture content are produced on sym3. Among them, the chlorophyll content has the best accuracy at coif3, and its overall accuracy and kappa coefficient are 0.90 and 0.85, respectively, as shown in Figure 6a. The fresh weight moisture content classification accuracy of coif4, sym2 and sym7 is higher, the overall accuracy is 0.86, 0.86 and 0.86, and the kappa coefficient is 0.79, 0.79 and 0.80, respectively (Figure 6b). The dry weight moisture content produced high classification accuracy on bior4.4, db2, db4, db5 and sym3, especially bior4.4 showed the highest overall accuracy and kappa coefficient (0.90, 0.86) (Figure 6c). It should be noted that chlorophyll content, fresh weight moisture content and dry weight moisture content produced the lowest classification accuracy on bior3.3, bio2.2 and sym2, with the overall accuracy of 0.52, 0.43 and 0.38, and kappa coefficients of 0.34, 0.26 and 0.21, respectively. These results show that the models constructed by different biochemical components on

various mother wavelet bases show different accuracy. Chlorophyll content data produced more stable overall accuracy (0.52–0.90) than fresh weight moisture content and dry weight moisture content data (0.43–0.86, 0.39–0.90). The overall accuracy of the chlorophyll content data (0.52–0.90) is higher than that of the fresh weight moisture content (0.43–0.86) and dry weight moisture content (0.39–0.90). On the whole, the larch damage identification model constructed on each biochemical component can better classify healthy trees, early damaged trees and damaged trees, which can meet the requirements of classification accuracy. Since our model has many outcomes, the confusion matrix of each outcome can only be used as a reference for supplemental data.

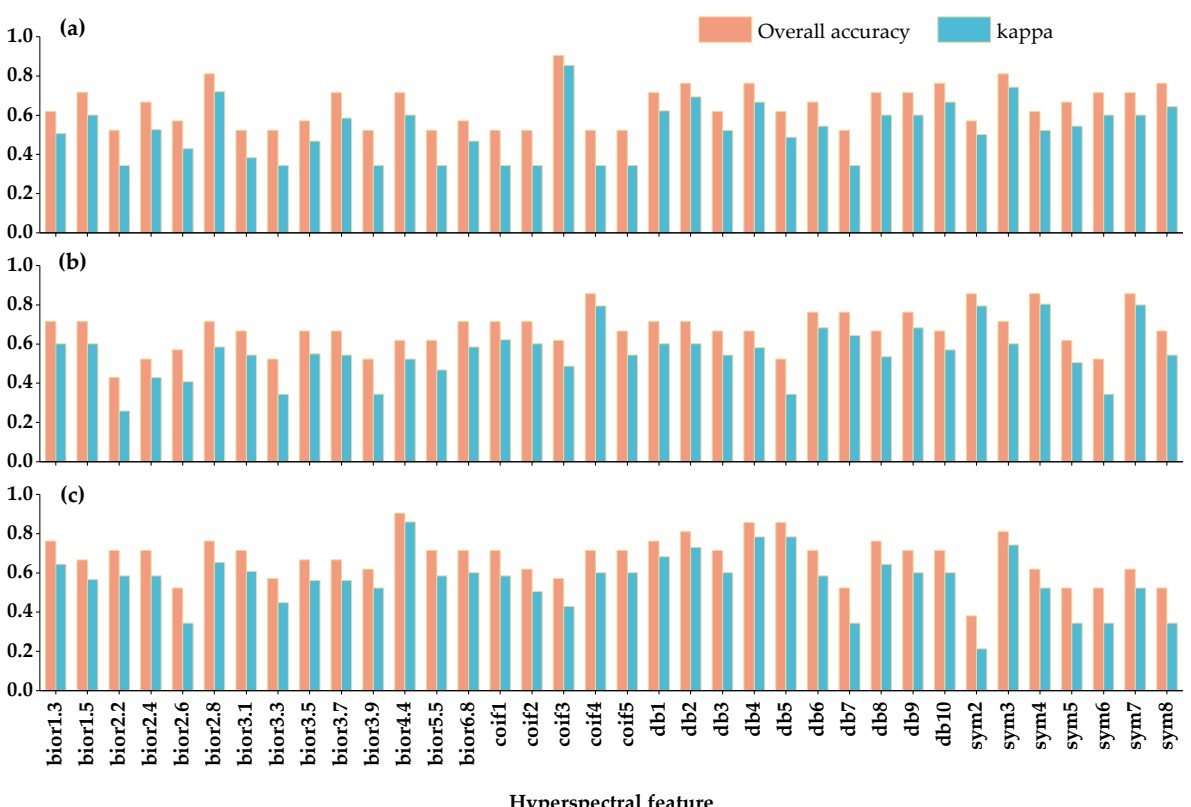

**Figure 6.** Accuracies of early pest discrimination models based on (**a**) chlorophyll content, (**b**) fresh weight moisture content, and (**c**) dry weight moisture content.

The confusion matrix and evaluation model index results of the optimal SVM classifier based on chlorophyll content, fresh weight moisture content and dry weight moisture content are shown in Tables 3–5. The overall accuracy was 90.48%, 85.71% and 90.48%, and the Kappa coefficients were 0.85, 0.79 and 0.86, respectively, which met the requirements of this study. The user accuracy and production accuracy of the healthy trees and the early damaged trees based on chlorophyll content were consistent, being 1 and 0.8, respectively. The user accuracy and production accuracy of the damaged forest were 91.68% and 100%, respectively. The user accuracy was 83.33%, 66.67% and 100%, and the production accuracy was 100%, 80% and 81.81%, respectively. The user accuracy of various sample trees based on dry weight moisture content was 100%, 71.43% and 100%, and the production accuracy was 80%, 100% and 90.91%, respectively. It can be seen that the identification rate of damaged trees was higher than that of healthy and early damaged trees, while the identification rate of early damaged trees is relatively low, especially the production accuracy of early damaged trees based on dry weight moisture content was only 71.43%. At the same time, it can also be seen that there are fewer cases of misclassification of injured trees into other categories, while there are relatively more cases of misclassification of healthy trees into early injured trees or early injured trees into healthy trees. This is because there is no

obvious quantitative change in the biochemical components of our healthy sample tree and the early victim sample tree, and there is a high gap between the healthy sample tree and the victim sample tree, which will affect the difficulty of classification between the early victim tree and the healthy tree. This was consistent with our experimental design. From the accuracy analysis of the above models, we can see that this experimental mode is workable to detect the damage status of larch forest in the early stage of inchworm invasion.

**Table 3.** Confusion matrix and evaluation index based on optimal mother wavelet basis (coif3) model of chlorophyll content.

| Classes | Healthy | Damaged Early | Damaged | Total | UA |
|---|---|---|---|---|---|
| OA: 90.47%, Kappa: 85.42% | | | | | |
| Healthy | 4 | 0 | 0 | 4 | 100% |
| Damaged Early | 1 | 4 | 0 | 5 | 80% |
| Damaged | 0 | 1 | 11 | 12 | 91.68% |
| Total | 5 | 5 | 11 | 21 | |
| PA | 80% | 80% | 100% | | 90.48% |

**Table 4.** Confusion matrix and evaluation index based on optimal mother wavelet basis (coif4) model of fresh weight moisture content.

| Classes | Healthy | Damaged Early | Damaged | Total | UA |
|---|---|---|---|---|---|
| OA: 85.71%, Kappa: 79.48% | | | | | |
| Healthy | 5 | 1 | 0 | 6 | 83.33% |
| Damaged Early | 0 | 4 | 2 | 6 | 66.68% |
| Damaged | 0 | 0 | 9 | 9 | 100% |
| Total | 5 | 5 | 11 | 21 | |
| PA | 100% | 80% | 81.82% | | 85.71% |

**Table 5.** Confusion matrix and evaluation index based on optimal mother wavelet basis (bior4.4) model of dry weight moisture content.

| Classes | Healthy | Damaged Early | Damaged | Total | UA |
|---|---|---|---|---|---|
| OA: 90.48%, Kappa: 86% | | | | | |
| Healthy | 4 | 0 | 0 | 4 | 100% |
| Damaged Early | 1 | 5 | 1 | 6 | 71.43% |
| Damaged | 0 | 0 | 10 | 10 | 100% |
| Total | 5 | 5 | 11 | 21 | |
| PA | 80% | 100% | 90.91% | | 90.48% |

## 4. Discussion

### 4.1. Sensitive Hyperspectral Feature Bands

This study proved that there was a good sensitivity between biochemical components and hyperspectral continuous wavelet coefficients in the larch forest under the stress of JLI in the whole hyperspectral band (Table 6). In some specific band ranges, spectral continuous wavelet coefficients can capture some spectral absorption and reflection characteristics corresponding to changes in the chlorophyll content and water content of leaves. Hence, when we use Pearson correlation to analyze the relationship between the hyperspectral continuous wavelet coefficient and the chlorophyll content and water content of leaves, we find that there are many sensitive band regions of different degrees. Some recent study also confirmed that hyperspectral continuous wavelet coefficients are sensitive to changes in plant biochemical parameters [19,53,54]. Such a large number of sensitive bands extracted will affect the stability and accuracy of later modeling because it contains a large number of continuous bands. We select the sensitive band corresponding to the

peak of the correlation coefficient by introducing Findpeaks function. The selected band has high sensitivity and ensures the dispersion of the band. We have identified different sample tree classes based on the sensitive bands extracted by Findpeaks function, but the results are not ideal. In addition, we find that the sensitive bands extracted based on Findpeaks function have multicollinearity problem, which reduces the stability of the model to some extent. In order to solve the problem of band multicollinearity, we use SPA function to further screen the highly sensitive spectral features, which can improve the speed and stability of later modeling. The SPA function is widely used in sensitive hyperspectral feature extraction [55–57]. After the Pearson–Findpeaks–SPA process, we extracted the sensitive hyperspectral bands that met our experimental requirements. These extracted sensitive hyperspectral bands can effectively capture the spectral reflection and absorption characteristics caused by differences in biochemical components, but they cannot correctly explain the contribution of the relevant bands to target detection. This is a shortcoming of this experiment. In addition, the main purpose of our experiment is to explore a new method to extract sensitive hyperspectral features. To establish an effective identification model of different degrees of damage of JLI infected larch forest. Therefore, the sensitive hyperspectral feature extraction experiment of Pearson–Findpeaks–SPA has certain application value.

**Table 6.** Sensitivity analysis results obtained by using Pearson–Findpeaks–SPA.

| Biochemical Component | Main Sensitive Band Range (nm) | Sensitive Band Etracted by Findpeaks–SPA (nm) |
|---|---|---|
| Chlorophyll content | 540–581, 596–699, 723–804, 954–956, 974–1011, 1134–1143, 1166–1199, 1245–1275, 1313–1386, 1412–1467, 1483–1485, 1500–1664, 1743–1770, 1784–1785 | 450, 644, 1042(coif3) |
| Fresh weight moisture content | 537–583, 595–709, 723–804, 953–962, 967–1010, 1053–1083, 1130–1139, 1162–1206, 1247–1279, 1315–1386, 1404–1468, 1480–1486, 1503–1672, 1746–1772, 1783–1785 | 446, 563, 1221, 1620, 1362, 1711(coif4) |
| Dry weight moisture content | 542–579, 598–679, 723–803, 978–1007, 1168–1195, 1245–1274, 1316–1386, 1416–1426, 1439–1465, 1503–1519, 1533–1654, 1744–1769, 1784–1785 | 382, 734, 951(bior4.4) |

*4.2. Future Trends and Prospects of Remote Sensing Monitoring of JLI Outbreak*

In this study, hyperspectral data were used to detect the damage of larch forest under the infection of JLI. This method has good stability in the identification of stressed forest. However, there are still several shortcomings. For example, we cannot determine the optimal pest index, because our sensitivity analysis results showed that although there was a good sensitivity between the biochemical components of the influence of the JLI and the spectral continuous wavelet coefficient, there was little difference in sensitivity between them. At the early stage of larch forest destruction, the canopy was green, and pathogens or other climatic conditions probably caused the decrease in chlorophyll content and water content in leaves. Therefore, our experiments cannot accurately judge the external factors of forest destruction. As a result, pathogen infection or climatic conditions may have damaged some trees in our study area to some extent before our data collection. This damage is likely to work with the JLI pest to damage the trees. Accordingly, when we only detect pest stress through the spectral difference between asymptomatic samples and infected samples, it is often not comprehensive enough. However, this method is especially useful for pest control. If the aim is to trace the cause of symptoms, we suggest future studies should consider forest conditions prior to pest attack. Specifically, a certain number of sample trees were

selected, and then the gradual changes of forest spectral reflectance and some biochemical components in the time series from pupal stage to adult stage were recorded. Such spectral and biochemical data can also explore the cause of symptoms using the method proposed in this study.

At present, the pest of inchworm is only distributed in larch forest of Mongolia, but the prediction and control of this pest should be carried out further. If the pest is left unchecked, it is highly likely to explode and spread to neighboring countries, causing unpredictable damage. We hope that more international researchers will pay attention to and take part in forest remote sensing research on JLI infection. Since we only did this under the condition that a single pest (JLI) infected a single forest (larch forest), the applicability of other forest pests needs to be further verified. In addition, remote sensing monitoring research on this insect pest has not been widely concerned by international researchers, and many articles supported by experimental technology are lacking. Therefore, the technical theories and methods of this study are referenced from other plant diseases and insect pest articles. Of course, we are only designing a study using ground non-imaging hyperspectral remote sensing, and implementing the technology is the key and the focus of future development.

### 5. Conclusions

This study showed that hyperspectral features based on chlorophyll content, fresh weight water content and dry weight water content could detect JLI infected larch forests. Nondestructive testing of healthy, early and damaged trees is important for controlling JLI outbreaks. We make full use of the full band spectrum of all sample trees and study the spectral characteristics of 36 parent wavelet bases obtained by continuous wavelet transform. A total of 11 mother wavelet bases (bior2.8, coif3, sym3, coif4, sym2, sym7, bior4.4, db2, db4, db5 and sym3) are considered to reveal the important characteristics of JLI infection. In addition, coif3, coif4 and bior4.4 are the optimal hyperspectral characteristics of chlorophyll content, fresh weight moisture content and dry weight moisture content in turn. The overall accuracy of the model based on these hyperspectral mother wavelet bases is 0.86–0.90, the kappa coefficient is 0.79–0.86, and the identification accuracy of early victims is more than 80%. These results prove the feasibility of extracting sensitive bands by Pearson–Findstacks–SPA and the practicability of SVM classification algorithm. This design method is helpful to JLI's early warning of Larch Forest stress and provides a very important theoretical basis and technical guidance for the detection of other large-scale pest outbreaks, which has a certain reference value.

**Author Contributions:** G.X. analyzed the data and wrote the paper; X.H. conceived and designed the experiments; G.D., T.N., A.D., M.A. and D.E. executed the experiments and measured the data; Y.X., B.G. and Y.B. revised the manuscript. All authors have read and agreed to the published version of the manuscript.

**Funding:** This study was supported by the National Natural Science Foundation of China (41861056, 61631011), Inner Mongolia Autonomous Region Science and Technology Plan Project (2021GG0183) and the Introduction of high levels talents to start scientific research projects of Inner Mongolia Normal University (2020YJRC051).

**Institutional Review Board Statement:** Not applicable.

**Informed Consent Statement:** Not applicable.

**Data Availability Statement:** Not applicable.

**Conflicts of Interest:** The authors declare no conflict of interest.

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
