# Peer review of "Detection of Larch Forest Stress from Jas’s Larch Inchworm (Erannis jacobsoni Djak) Attack Using Hyperspectral Remote Sensing"

_remotesensing, doi:10.3390/rs14010124_

Round 1

Reviewer 1 Report

Dear. Authors,

Thanks for your revisions.
I checked your resubmission files.
The contents of manuscript have been improved. Thank you.

However, regarding the importance analysis of features in SVM, more consideration would be needed.

As I gave a comment, OA and Kappa in Table 4-6, and Figure 7 are not enough to discuss the priority of hyperspectral feature used in this study.
Because, one of the major objectives of this study is to discuss 'which compositions of hyperspectral wavelet bands are useful to detect the JLI pest?'. And authors kindly replied to the my previous comment as follows: 'the SVM model was only chosen and it didn't have any function to investigate the importance of input variables'. 

Regarding this answer, I have a question.
Are you using "python" for radial basis function (RBF) kernel?

  • If you use python for LIBSVM RBF kernel, you can use "sklearn.inspection.permutation_importance" for visualizing variables importance, easily.
    There are also many other options in attributes of python for checking variables importance when the SVM was built (i.e., "feature_importances_", "coef_", and "shap.KernelExplainer").
  • If you use matlab, 'fscnca' function will be helpful to show relevant features for your classification.

Please, consider applying some of those methods. 
If the analysis was successfully done, please, submit it again for the next revision round.
I encourage you to try some of the above mentioned methods for identifying important variables for your SVM model.

Then, I will give a decision for this paper after I check the resubmitted manuscript.

Good luck!

Author Response

Response: We would like to express our sincere thanks to you for the constructive and positive comments. In particular, you have provided me with solutions and functions for the problem, for which I am grateful again. But after careful consideration, due to the initial design of this article, we are temporarily unable to implement it according to your situation. I want to explain our point of view. In this article, we have extracted the sensitive hyperspectral band through the data processing process of Pearson-Findstacks-SPA. The sensitive hyperspectral bands that are finally screened in layers are equivalent to the important input variables of the SVM model. As shown in Figure 5(a), in the sensitive hyperspectral band based on the extraction of chlorophyll content, only one or only a few bands are extracted from some mother wavelet bases. For example, only one sensitive band of mother wavelet bases such as bior6.3, bior5.5, db3 and db9 is extracted, and most of the sensitive bands extracted from other mother wavelet bases are only in the range of 2-5. If we further analyze the priority of important bands on this basis, we think it is unnecessary. In the article, we used three indicators (chlorophyll, dry weight moisture content and fresh weight moisture content) and 36 mother wavelet bases. Obviously, due to the characteristics of the data, further importance analysis (3*36=108 results) will also bring some difficulty to our analysis. If we follow your thoughts, it is not necessary especially for some mother wavelet bases that only extract a one sensitive band. Therefore, importance analysis of SVM features may be an unnecessary process for the design of this paper. Hope you can understand the purpose of our experiment design. If there are any other modifications we could make, we would like very much to modify them and we really appreciate your help. We appreciate for your warm work earnestly, and hope that the corrections will meet with approval. We deeply appreciate the Editor’s and Reviewers’ thoughtful input, and we hope that the revised manuscript will meet with your approval. Once again, thank you very much for your valuable comments and suggestions.

Reviewer 2 Report

The aim of this study is to investigate the possible use of SVM to detect untouchable pest symptoms in forests using hyperspectral reflectance and biochemical components data. The results of this study showed the effectiveness f using SVM for this aim due to its high accuracy in predicting the Chl content, fresh weight moisture, and dry weight moisture content. The idea behind this study was attractive, and it may bring attention. However, some minor changes need to be addressed before publishing in this journal:

 Please provide more explanation about the nature of the algorithms used in this study. There is very limited information about the SVM in the introduction.

Remove Figure 3 as it is not necessary to be in the main body of the paper.

Provide more explanation in the figure title, specifically figure 4. The title of each figure should be self-explanatory.

Why did you not estimate the F1, precision, and recall? If you did, please include and explain them in the body of the paper.

Line 332, do you mean coefficient of determination?

Author Response

We have carefully taken into account the reviewers’ comments and amended the manuscript accordingly. We have revised the manuscript in accordance with the comments and marked all the amends on our revised manuscript. The revised manuscript was marked with red color and the responses were presented in blue text.

If there are any other modifications we could make, we would like very much to modify them and we really appreciate your help. We appreciate for your warm work earnestly, and hope that the corrections will meet with approval. We deeply appreciate the Editor’s and Reviewers’ thoughtful input, and we hope that the revised manuscript will meet with your approval. Once again, thank you very much for your valuable comments and suggestions.

Round 2

Reviewer 1 Report

Dear. Authors,

Thanks for your revision and kind reply.
I hope that your analysis of Jas’s Larch Inchworm with tremendous data could be helpful for further studies regarding this theme.

Have a nice holiday,

This manuscript is a resubmission of an earlier submission. The following is a list of the peer review reports and author responses from that submission.

Round 1

Reviewer 1 Report

see the attachment.

Reviewer 2 Report

PROSPECT model are widely used in dealing with hyperspectral data for understanding the different experiments at leaf level (for LAI, water content, chlorophyll content) in this case is the detection of infestation. Could you include some information on radiative transfer nature of PROSPECT model in the introduction and in your discussion section to support your findings about the detection of infestation. I would also recommend to include PROSPECT model simulation to compare with your present results in this MS or in your future project as will significantly improve your paper and the journal. 

Reviewer 3 Report

This paper is describing hyperspectral characteristics of the stress of Jas's larch inchworm. Machine learning approaches (e.g., RF and SVM) were used for identifying the stage of pest infestation. In this study, feature selection results in between continuous wavelet coefficients of collected hyperspectral data have been provided.

I think the paper quality is average. However, some parts (introduction, study area, methodology, and results) need to be revised for further process of publication as an original article in the journal of remote sensing.

Major comments:

  1. Introduction needs to be revised by adding,
    1) the importance of larch forest in terms of climate changes,
    2) more specific description regarding references [10 - 14] (line 77) (Authors also did a ground survey. That is, still, traditional technique is necessary for modeling it, isn't it? Please, show us another motivation or importance to develop new system),
    3) the reason why 'detection' is so important in Mongolia (e.g., economic reasons, biodiversity reasons, regarding the fatal ratio & infection speed, and so on),
    4) the reason why 'early' detection is so important and how could be the early stage prediction available by using the proposed method from this study (i.e., RF, SVM)?
    - These contents were expected to appear in Introduction part, however I couldn't find it.
  2. Objectives (1) and (2) can be merged into one.

  3. Introduction includes not relevant information with this topic. I suggest to 1) remove Lines 57 - 60 and Lines 68 - 70. Because those are not directly related to 'the larch inchworm pest', but includes too elementary information in remote sensing research field. 

  4. Line 100, Sorry, I still can not understand what is the relation between this study and 'early' detection.. since this study doesn't include any time-series analysis.. Could you explain it in here?

  5. Materials and Methods
    1) Line 106, Suggest to change 'average temperature in June and July (x) --> annual average temperature (o) (better to understand)'
    2) Lines 107 - 109, add citations which can support this part. It seems to be important.
    3) Line 107, about annual precipitation of 200-3000mm --> what did you refer to, for this number? Please, add a citation and baseline durations (how many years were considered).

  6. In '2.2.2 Hyperspectral data collection and preprocessing', line 182.
    Not only in here, but also in some other parts, authors mention 'to improve accuracy'. I feel it sounds tricky. I suggest to use - 'to improve universality' or 'to develop a robust model' and so on.

  7. Lines 192-202, why is water content related with inchworm? Please, add a physical meaning to select fresh and dry water content as the input variables for expressing the pest stress into the Introduction.

  8. Figure 4. Hard to understand. Some processes are not connected to their input data, but output is connected. Please, revise this flowchart.
  9. Lines 216 - 222, Includes generally well-known information. Can be removed.
  10. Lines 263 - 271, too lengthy. Please, it needs to be compact.

  11. 3.3 Model results. 
    1) Lines 362 - 383. Please, show us the error matrix with optimal cases.
    2) Figure 6. Color legend is not suitable
    3) Figure 6. In scatter plots of 'RF-OA vs SVM-OA', RF show no changes in values. What happened?
    4) Figure 6. No description with '+' symbol.
    5) Table 2. Optimal features, OA, and Kappa have been shown. This is not enough to analyze the results from RF and SVM. Usually, Gini index, importance of input variables are used. Add the analysis regarding variable importance.

  12. Discussion 
    1) Lines 422-423, please add sentences for supporting potential use of NIR.
    2) Lines 510 - 514, add citations.
    3) Lines 520 - 521, and 527 - 529. Need to be included to Introduction.

Minor comments:

13. Line 125, please, explain what is 'a total of 6 typical branches'.

14. Lines 314 - 322. Citations?

15. Figure 4. What is the meaning of small ticks in left y-axis? Please, add description of this y-ticks in the figure caption.

16. Line 461. Check '21-10'.

17. Reference format check carefully, please.

Reviewer 4 Report

The authors provide their methodology in achieving early detection of stress in a single species forest with the use of hyperspectral data. The manuscript has been apparently improved since its first submission and can be of interest to the readers. Overall it has a good quality and clear structure and the experiment is sound. The results are promising although additional research is required, according to the authors. 

Some minor corrections: 

  • P.3, line 105: should change hm to km2 ?
  • P.8, line 286: Should be figure 4 instead of 2